# Decoupling Weighing and Selecting for Integrating Multiple Graph Pre-training Tasks

**Tianyu Fan**[1,2,*], **Lirong Wu**[1,2,*], **Yufei Huang**[1,2], **Haitao Lin**[1,2],
**Cheng Tan**[1,2], **Zhangyang Gao**[1,2], **Stan Z. Li**[1,†]
[1]Westlake University, [2]Zhejiang University
{fantianyu,wulirong,huangyufei,linhaitao}@westlake.edu.cn;
{tancheng,gaozhangyang,stan.zq.li}@westlake.edu.cn

## Abstract

Recent years have witnessed the great success of graph pre-training for graph representation learning. With hundreds of graph pre-training tasks proposed, integrating knowledge acquired from multiple pre-training tasks has become a popular research topic. In this paper, we identify two important collaborative processes for this topic: (1) *select*: how to select an optimal task combination from a given task pool based on their compatibility, and (2) *weigh*: how to weigh the selected tasks based on their importance. While there currently has been a lot of work focused on weighing, comparatively little effort has been devoted to selecting. This paper proposes a novel instance-level framework for integrating multiple graph pre-training tasks, Weigh And Select (WAS), where the two collaborative processes, *weighing* and *selecting*, are combined by decoupled siamese networks. Specifically, it first adaptively learns an optimal combination of tasks for each instance from a given task pool, based on which a customized instance-level task weighing strategy is learned. Extensive experiments on 16 graph datasets across node-level and graph-level downstream tasks have demonstrated that by combining a few simple but classical tasks, WAS can achieve comparable performance to other leading counterparts. The code is available at https://github.com/TianyuFan0504/WAS.

## 1 Introduction

Relationships between entities in various real-world applications, such as social media, molecules, and transportation, can be naturally modeled as graphs. Graph Neural Networks (GNNs) (Hamilton et al., 2017; Veličković et al., 2017; Wu et al., 2023a;e; 2022c) have demonstrated their powerful capabilities to handle relation-dependent tasks. However, most of the existing work in GNNs is focused on supervised or semi-supervised settings, which require labeled data and hence are expensive and limited. Recent advances in graph pre-training (Wu et al., 2021; Xie et al., 2021; Liu et al., 2021c) are aimed to reduce the need for annotated labels and enable training on massive unlabeled data. The primary purpose of graph pre-training is to extract informative knowledge from massive unlabeled data and the learned knowledge can then be transferred to some specific downstream tasks. While hundreds of graph pre-training tasks have been proposed in this regard (Sun et al., 2019; Hu et al., 2020b; Zhu et al., 2020b; You et al., 2020a; Zhang et al., 2020; Wu et al., 2023d), as shown in Fig. 1(a), there is no single pre-training task that performs best for all datasets.

Therefore, integrating or more specifically linearly weighing multiple tasks, has emerged as a more effective approach than designing more complex tasks. For example, AutoSSL (Jin et al., 2021) combines the weights of task losses based on a pseudo-homophily measure, and ParetoGNN (Ju et al., 2022) reconciles pre-training tasks by dynamically assigning weights that promote the Pareto optimality. Another related work is AUX-TS (Han et al., 2021), which also adaptively combines different tasks, but this combination appears in the fine-tuning stage. However, all three works perform task weighing in a global manner, ignoring the fact that different instances (e.g., nodes in a social network or graphs in a molecular dataset) may have their own specificities. To solve this, AGSSL (Wu et al., 2022a) has been proposed to learn instance-level task weighing strategies during the fine-tuning stage. Nonetheless, these works to *weigh* all tasks focus only on the **importance** issue, but ignore the **compatibility** issue, i.e., the possible conflicts between different tasks, which cannot be resolved by simply weighing all tasks. More seriously, as the task pool expands, **compatibility** issue

---

*: Equal Contribution. †: Corresponding Author.

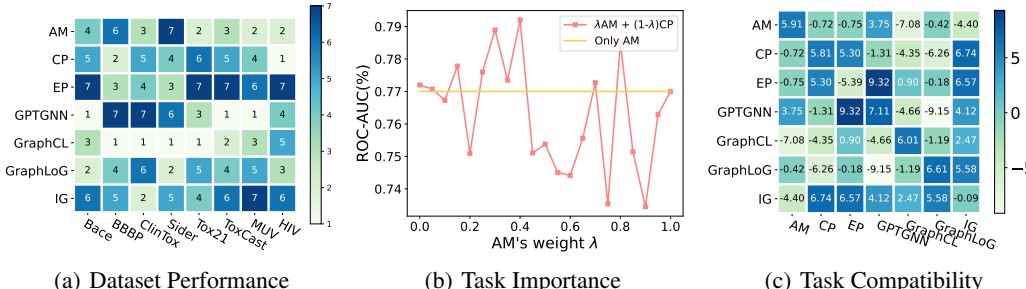

(a) Dataset Performance          (b) Task Importance          (c) Task Compatibility

Figure 1: **(a)** Performance ranking (1: best, 7: poorest) of seven pre-training tasks (rows) on eight datasets (columns). **(b)** Performance fluctuation on `Bace` (molecule dataset) when combining two tasks, `AM` and `CP`, with different task weight $\lambda$. **(c)** Performance gains or drops over that without pre-training when combining two tasks (diagonal represents only a single task) on `Bace`.

Table 1: A comprehensive comparison between previous methods and ours. *Stage* indicates at what stage the method is applied. *Task Type* represents the levels of tasks that the method can handle.

|  | Instance-level | Weighing | Selecting | Task Type | Stage |
|---|:---:|:---:|:---:|:---:|:---:|
| AutoSSL (Jin et al., 2021) | ✗ | ✓ | ✗ | Node | pre-training |
| ParetoGNN (Ju et al., 2022) | ✗ | ✓ | ✗ | Node | pre-training |
| AUX-TS (Han et al., 2021) | ✗ | ✓ | ✗ | Node | fine-tuning |
| AGSSL (Wu et al., 2022a) | ✓ | ✓ | ✗ | Node&Graph | fine-tuning |
| WAS (ours) | ✓ | ✓ | ✓ | Node&Graph | fine-tuning |

becomes more severe, which deprives existing methods of the ability to keep gaining performance growth. Therefore, it is necessary to *select* some tasks to solve it. In addition, previous works have only evaluated their effectiveness on node-level tasks but neglected graph-level tasks. We have summarized the properties of these methods in Table. 1 and compared them with WAS.

We would like to raise several issues through investigation on several classical graph pre-training tasks, including AttrMask (AM), ContextPred (CP) (Hu et al., 2020b), EdgePred (EP) (Hamilton et al., 2017), GPT-GNN (Hu et al., 2020c), GraphCL (You et al., 2020a), GraphLoG (Xu et al., 2021), and InfoGraph (IG) (Sun et al., 2019), as shown in Fig. 1. Let us first draw an interesting conclusion: both *importance weighing* and *task selecting* of tasks are quite important, where the former addresses the issue of task **importance**, while the latter addresses the issue of task **compatibility**.

Fig. 1(b) compares the performance under varying importance weights $\lambda$ when combining AM and CP. We can see that the best performance is achieved at $\lambda = 0.4$. While task weighing focuses on the importance of each task, it neglects compatibility between different tasks, which limits existing methods from achieving higher performance. The results in Fig.1(c) illustrate the huge impact of the compatibility issue when integrating multiple tasks. It can be seen that not all combinations can bring performance improvements, and some of them even degrade performance (combination of AM and GraphCL brings a -7.08 drop), which highlights the necessity of selecting suitable task combinations. Note that Fig.1(c) shows only the combination between two tasks. As the task pool expands, such combinations become more complex, and conflicts between different tasks may become more severe.

Based on the above investigations, here we would like to ask: *How to address the importance and compatibility issues between different tasks when integrating them during the fine-tuning stage?* We identified two key issues in the combining process: (1) *task selecting* – how to select an optimal combination from a given task pool based on the **task compatibility**, (2) *importance weighing* – how to weigh the **importance** of the selected tasks. These two are obviously related or collaborative. More important tasks should be selected more, but selecting tasks based solely on importance can lead to severe task conflicts. While previous works, AutoSSL, ParetoGNN, AUX-TS, and AGSSL, have focused on *importance weighing*, they have all overlooked *task selecting*, which has deprived them of the ability to keep gaining performance growth as the task pool grows larger.

In this paper, we propose a novel framework, *Weigh And Select* (WAS) for task selecting and importance weighing. The two collaborative processes are combined in decoupled siamese networks, where (1) an optimal combination of tasks is selected for each instance based on a sampling distribution calculated based on the task compatibility, and (2) task weights are then calculated for the selected tasks according to their importance. To the best of our knowledge, this work is the first attempt to use the **weighing & selecting** strategy for integrating multiple graph pre-training tasks. Extensive

experiments on 16 datasets show that WAS can achieve comparable performance to other leading counterparts for both node-level and graph-level tasks. Our contributions are summarized as follows: (1) We show the limitations of current *weighing*-only schemes and demonstrate the importance of *task selecting* process. (2) To the best of our knowledge, we are the first to identify two important collaborative processes: *select* and *weigh*; we provide extensive experiments to explain in detail why the collaboration of the two is important, how it differs from the *weighing*-only based methods and why do the two processes need to be decoupled. (3) We propose a novel framework to adaptively, dynamically, and compatibly select and weigh multiple pre-training tasks for each instance separately.

## 2 RELATED WORKS

**Graph Pre-Training.** Graph neural networks (GNNs) are powerful tools to capture useful information from graph data (Zügner & Günnemann, 2019; Wu et al., 2022d;b; Hu et al., 2020a). Recently, there have been lots of efforts in pre-training GNNs to alleviate the need for expensive labeled data and improve the generalization ability. These methods usually use various well-designed pre-training tasks to pre-train GNNs on large unlabeled datasets. Generally, mainstream pre-training tasks can be divided into three categories: generative, contrastive, and predictive. The generative methods, such as EdgePred (Hamilton et al., 2017), AttrMask (Hu et al., 2020b), and GPT-GNN (Hu et al., 2020c), focus on reconstructing important information for each graph at the intra-data level. Besides, the contrastive methods, such as ContextPred (Hu et al., 2020b), GraphLoG (Xu et al., 2021), GraphCL (You et al., 2020a), and JOAO (You et al., 2021), apply data transformations to construct different views for each graph, aiming to learn representations to distinguish the positive views from the negative views. The predictive methods, such as G-Motif (Rong et al., 2020), CLU (You et al., 2020b), and PAIRDIS (Jin et al., 2020), generally self-generate labels by some simple statistical analysis and then perform prediction-style tasks. More details can be found in **Appendix A**.

**Multi-Tasking Learning.** There are many works on multi-tasking learning (Doersch & Zisserman, 2017; Ren & Lee, 2017; Zamir et al., 2018; Wu et al., 2020; Yu et al., 2020; Wang et al., 2022) on CV or NLP domains. Due to the immense success of Transformer (Vaswani et al., 2017), many methods from non-graph domains are applicable only to the transformer architecture (He et al., 2022; Zhu et al., 2023; Wang et al., 2022). In addition, lots of tasks focus on designing methods to combine losses (Yu et al., 2020; Liu et al., 2021a), ignoring the powerful potential of instance-level design. The existing methods on graphs can be broadly classified into two categories, global-level and instance-level. For global-level, AUX-TS (Han et al., 2021) combines different tasks to promote a target pre-training task's performance. AutoSSL (Jin et al., 2021) combines the weights of losses on tasks by measuring pseudo-homophily, and ParetoGNN (Ju et al., 2022) reconcile tasks by dynamically assigning weights that promote the Pareto optimality. They trained the model by combining losses, ignoring the different needs of different instances. For instance-level, AGSSL (Wu et al., 2022a), which is closest to us, designs its weighing function to approach an ideal Bayesian teacher for each instance. Despite their great success, all the above methods focus only on **importance weighing** but ignore **task selecting**.

## 3 PRELIMINARY

**Notations.** Let $\mathcal{G} = (\mathcal{V}, \mathcal{E})$ denote a graph, where $\mathcal{V} = \{v_1, v_2, \cdots, v_N\}$ and $\mathcal{E} \subseteq \mathcal{V} \times \mathcal{V}$ denote the node set of $|\mathcal{V}| = N$ nodes and the edge set. Given a set of graphs $G = \{\mathcal{G}_1, \mathcal{G}_2, \cdots, \mathcal{G}_M\}$, graph classification aims to learn a GNN encoder $h_\theta(\cdot) : \mathcal{G} \to \mathbb{R}^F$ and an additional projection head $g_\omega(\cdot) : \mathbb{R}^F \to \mathbb{R}^C$ to adapt to downstream tasks, where $C$ is the number of category.

**Pre-training and fine-tuning on Graphs.** Graph pre-training aims to extract informative knowledge from massive unlabeled data $\mathcal{D}_{\text{pre}}$ through a pre-training task $\mathcal{L}_{\text{pre}}(\theta)$ and then transfer the learned knowledge to a downstream task $\mathcal{L}_{\text{down}}(\theta, \omega)$ on the labeled data $\mathcal{D}_{\text{down}}$, which includes two steps:

(1) Pre-training a GNN encoder $h_\theta(\cdot)$ on an unlabeled dataset $\mathcal{D}_{\text{pre}}$, with the objective formlulated as

$$\theta_{\text{pre}} = \arg\min_\theta \mathcal{L}_{\text{pre}}(h_\theta; \mathcal{D}_{\text{pre}}). \tag{1}$$

(2) Fine-tuning the pre-trained GNN encoder $h_{\theta_{\text{pre}}}(\cdot)$ with a prediction head $g_\omega(\cdot)$ on the labeled dataset $\mathcal{D}_{\text{down}}$, defined as

$$\min_{(\theta, \omega)} \mathcal{L}_{\text{down}}(h_{\theta_{\text{pre}}}, g_\omega; \mathcal{D}_{\text{down}}). \tag{2}$$

**Multi-teacher Knowledge Distillation.** Given $K$ teacher models $f_1^{\mathcal{T}}(\cdot), f_2^{\mathcal{T}}(\cdot), \cdots, f_K^{\mathcal{T}}(\cdot)$ and a student model $f^{\mathcal{S}}(\cdot)$, the multi-teacher knowledge distillation extracts knowledge from multiple

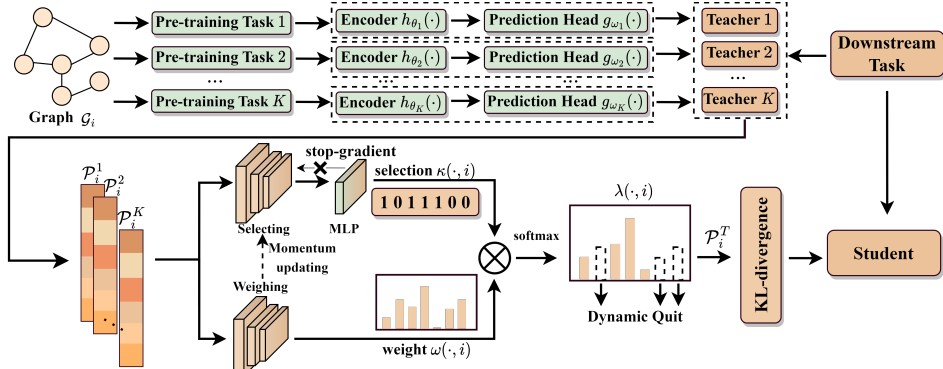

Figure 2: **Overall workflow of WAS.** Firstly, we train multiple teachers with different pre-training tasks. Secondly, we pass the teacher's representations to two modules (Selecting and Weighing) to get the selecting results $\kappa(\cdot, i)$ and initial weights $\omega(\cdot, i)$ for each instance $\mathcal{G}_i$. Finally, we weigh only those selected teachers to get weights $\lambda(\cdot, i)$ and distill the integrated distributions into the student.

teachers and then distills the extracted knowledge into a student, which can be formulated as follows

$$\mathcal{L}_{MKD} = \mathcal{L}_{KL}\left(\sum_{k=1}^{K} \lambda_k f_k^{\mathcal{T}}(\mathcal{G}), f^{\mathcal{S}}(\mathcal{G})\right), \tag{3}$$

where $\lambda_k$ is the weight of $k$-th teacher that satisfies $\sum_{i=1}^{K} \lambda_k = 1$, and $\mathcal{L}_{KL}(\cdot)$ denotes the Kullback–Leibler divergence that measures the distribution differences.

## 4 METHODOLOGY

The graph pre-training tasks are designed to provide more information to the model during pre-training, and in this sense, using more tasks can provide richer information. However, there are two issues to be addressed when combining tasks, namely, the **importance** of each task and the **compatibility** between tasks. Since different tasks exhibit different importance and compatibility on different data, in this paper, we propose a novel instance-level framework, _Weigh And Select_ (WAS), as shown in Fig. 2, which can be divided into two steps: knowledge extraction and knowledge transfer. In the knowledge extraction step, we train multiple teachers with different tasks to extract different levels of knowledge. In the knowledge transfer step, we integrate those knowledge for each instance by weighing and selecting, and then distill the integrated knowledge into the student model. To deal with the **importance** and **compatibility** issues, we design decoupled siamese networks with stop-gradient and momentum updating, which assigns different weights to each teacher and determine whether they should be selected, and then only weigh those selected teachers, i.e., _weigh and select_.

In this section, we delve into the details of our framework WAS by answering the following questions:

**Q1. (Instance):** How to design a framework that enables customized schemes for each instance?

**Q2. (Importance):** How to design the weighing module to address the importance issue?

**Q3. (Decouple):** How to decouple selecting from weighing to prevent their results from overlapping?

**Q4. (Compatibility):** How to design the selecting module to address the compatibility issue?

### 4.1 INSTANCE-LEVEL MULTI-TEACHER KNOWLEDGE DISTILLATION

To answer **Q1 (Instance)**, we adopt multi-teacher knowledge distillation (MKD) to achieve knowledge transfer in our framework. MKD was first proposed to obtain a model with fewer parameters, i.e., model compression (Wu et al., 2023b;c), but here we use it for distilling the knowledge of different teachers into one student model. As opposed to the method of combining different losses, used in AutoSSL (Jin et al., 2021) and ParetoGNN (Ju et al., 2022), MKD can get the distribution of each teacher on each individual instance. This means that the teacher's impact on students is independent at the instance-level, so we are able to learn customized teacher combinations for each instance to enable the student model to learn a better representation. Furthermore, since we directly weigh the output distributions of teachers rather than the task losses, the learned weights can truly reflect the importance of different teachers, because their output distributions are at almost the same level for a given instance. However, different losses may be on different orders of magnitude, so the weights of different losses cannot directly reflect the importance of different graph pre-training tasks.

To combine different teachers at the instance-level, we learn customized weighing and selecting strategies for each instance separately. Specifically, given $K$ teacher models $f_1^{\mathcal{T}}(\cdot), f_2^{\mathcal{T}}(\cdot), \cdots, f_K^{\mathcal{T}}(\cdot)$ trained by different pre-training tasks, our goal is to obtain an ideal combination of them for each instance $\mathcal{G}_i$. The whole framework can be divided into three key steps: (1) Get the label distribution $\mathcal{P}_i^k = f_k^{\mathcal{T}}(\mathcal{G}_i)$ of the $k$-th teacher $f_k^{\mathcal{T}}(\cdot)$ on $i$-th instance $\mathcal{G}_i$. (2) Pass the obtained distributions through two mutually independent modules $\mathbb{W}$ and $\mathbb{S}$, i.e., weighing (introduced in 4.2.1) and selecting (introduced in 4.2.3), to obtain initial weight $\omega(k,i) \in (0,1]$ and selecting results $\kappa(k,i) \in \{0,1\}$, and then weigh only those selected teachers by softmax to output teacher weight $\lambda(k,i) \in (0,1]$. (3) Integrate the outputs of different teachers to obtain an integrated teacher distribution $\mathcal{P}_i^T$ as follows,

$$\mathcal{P}_i^T = \sum_{k=1}^{K} \kappa(k,i)\lambda(k,i)\mathcal{P}_i^k, \tag{4}$$

where $\sum_k \kappa(k,i)\lambda(k,i) = 1$, and then distill the integrated distribution $\mathcal{P}_i^T$ into the student $f^{\mathcal{S}}(\cdot)$ via multi-teacher knowledge distillation, with the learning objective formulated as

$$\min_{\theta,\omega,\lambda,\kappa} \mathcal{L}_{\text{down}}(\theta,\omega) + \alpha \sum_{i=1}^{M} \mathcal{L}_{KL}\left(\mathcal{P}_i^T, \mathcal{P}_i^S\right). \tag{5}$$

## 4.2 SIAMESE NETWORKS FOR TASK WEIGHING AND SELECTING

An instance-level framework has been presented in Sec. 4.1 to transfer knowledge from multiple teachers to a student, but it doesn't tell us how to weigh and select. In this subsection, we propose siamese networks that consist of two modules dealing with **importance** and **compatibility**.

### 4.2.1 TASK WEIGHING BY IMPORTANCE MODELING

Here, to answer **Q2 (Importance)**, we design a weighing module $\mathbb{W}$ to adaptively learn suitable task weights for each instance. The first question is, how to model the importance issue. In some current multi-task learning methods (Jin et al., 2021; Ju et al., 2022), they directly optimize the weights $\lambda$ of different tasks (usually in the form of the coefficient of losses). However, this is only useful at the global-level as different tasks have different forms of loss. For example, GraphCL performs contrastive learning between graphs (graphs as instances), while AM masks node attributes on a single graph (nodes as instances). To solve this, we use a latent space described by the variable $\{\boldsymbol{\mu}_k\}_{k=1}^{K}$ and associate each teacher with a latent factor $\boldsymbol{\mu}_k \in \mathbb{R}^C$ that captures the local importance of different teachers. The importance weight of the $k$-th teacher to graph $\mathcal{G}_i$ can be calculated as follows:

$$\omega(k,i) = \frac{\exp\left(\zeta_{k,i}\right)}{\sum_{k=1}^{K} \exp\left(\zeta_{k,i}\right)}, \quad \text{where } \zeta_{k,i} = \boldsymbol{\nu}^T\left(\boldsymbol{\mu}_k \odot f^{\mathcal{S}}(\mathcal{G}_i)\right) \tag{6}$$

where $\boldsymbol{\nu} \in \mathbb{R}^C$ is a vector of global parameters to be learned, which determines whether the value of each dimension in $\left(\boldsymbol{\mu}_k \odot f^{\mathcal{S}}(\mathcal{G}_i)\right)$ has a positive impact on the importance.

After modeling the importance issue, the second question is, how to optimize the learning of $\omega(\cdot,\cdot)$. Inspired by AGSSL (Wu et al., 2022a), an optimal weighing scheme should make the integrated teacher distribution $\mathcal{P}^T(\mathcal{G})$ close to the Bayesian teacher $\mathcal{P}^*(\mathcal{G})$, which provides true class probabilities but is often unknown. Luckily, Menon et al. (2021) demonstrate that $\mathcal{P}^*(\mathcal{G})$ can be estimated using the ground-truth label $\mathbf{e}_y$. Therefore, we approximately treat $\mathcal{P}^*(\mathcal{G}) \approx \mathbf{e}_y$, optimize $\omega(\cdot,\cdot)$ by minimizing the binary cross entropy loss $\mathcal{L}_W = \frac{1}{|\mathcal{D}_{\text{down}}|} \sum_{i \in \mathcal{D}_{\text{down}}} \ell(\mathcal{P}^T(\mathcal{G}_i), \mathcal{P}^*(\mathcal{G}_i))$ on the downstream labeled set, and estimate weights $\{\omega(k,j)\}_{k=1}^{K}$ for any unlabeled instance $\mathcal{G}_j$.

### 4.2.2 DECOUPLE MODULES BY SIAMESE NETWORKS

Before going deep into the selecting module, we must first clarify one question: what kind of selecting results do we want? As mentioned earlier, the weight $\omega(k,i)$ can characterize the importance of each task, so it is natural to select those tasks that are important and remove those that are not, i.e., using the weight as the selecting probability. However, this scheme is equivalent to explaining **compatibility** in terms of **importance** and completely confuses the two, which defies our expectations. Fig. 3 shows the results of three selecting methods on four datasets. It can be seen that the completely importance-based selection

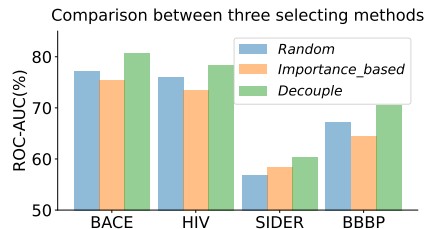

Figure 3: A detailed comparison of three task selecting schemes on four datasets.

method neglects compatibility and is sometimes even worse than random selection. However, it would be a pity to discard the task importance altogether, as it undoubtedly provides rich information that can be used to guide task selection, so we wanted to design a selecting module to utilize this information but not rely on it. That is, we want the selecting results and weights to be *related but not coupled with each other*. Then, the two modules should be architected in such a way that the selecting results can make use of the weights but are not directly derived from them. We call this process *decouple*. As shown in Fig. 3, the decoupled selection achieves the best performance.

To answer **Q3 (Decouple)**, we construct the selecting module and the weighing module in the form of siamese networks inspired by Nair & Hinton (2010) to get weights and selecting results simultaneously, and then weigh only those selected teachers. Such a construction can strictly distinguish between the two modules, allowing the selecting module to focus on solving the task **compatibility** problem and the weighing module to focus on the task **importance** problem.

### 4.2.3 Task Selecting by Quit Mechanism

To answer **Q4 (Compatibility)**, we propose a novel selecting module $\mathbb{S}$ to adaptively resolve the compatibility between tasks and select the most suitable task combination for each instance separately.

As mentioned in Sec. 4.2.2, we construct the selecting module and the weighing module in the form of siamese networks. However, since the two modules have the same architecture, if we update them in the same way (e.g., both by back-propagation), their outputs are highly likely to be re-coupled together. Therefore, we cut off the selecting module's gradient back-propagation and adopt the momentum updating to optimize the parameters of the selecting module $\mathbb{S}$. Let $\theta_{weighing}$ denote the parameters of weighing module $\mathbb{W}$ and $\theta_{selecting}$ denote the parameters of selecting module $\mathbb{S}$, the process of momentum updating can be written as follows:

$$\theta_{selecting} = m * \theta_{selecting} + (1 - m) * \theta_{weighing}, \tag{7}$$

where $m$ is the momentum updating rate, which controls the similarity of $\mathbb{S}$ and $\mathbb{W}$. This setup enables the selecting module to acquire enough knowledge from the historical weights of different teachers to guide the selecting results because $\theta_{selecting}$ is based on historical $\theta_{weighing}$.

However, since $\theta_{selecting}$ are all superimposed by $\theta_{weighing}$ at different epochs, the output of the selecting module is still likely to be close to the weighing module if $\mathbb{W}$ converges earlier. Therefore, we add an additional projection head (implemented as MLP) after $\mathbb{S}$ to enhance the discrepancy between the selecting and weighing results. Such an organization enables $\mathbb{S}$ to still utilize the information of weighing but be thoroughly decoupled from the weighing module, allowing the two modules to do their work more independently. Given the output $\kappa_S(k, i)$ of the selecting module (just like the output of the weighing module $\omega(k, i)$), we feed $\kappa_S(k, i)$ into the MLP and then normalize the result $\mathtt{MLP}(\kappa_S(k, i))$ to ensure that at least one teacher is selected for each instance, defined as

$$\kappa_{\mathrm{norm}}(k, i) = \frac{\mathtt{MLP}(\kappa_S(k, i))}{\max_{j=1}^{K} \mathtt{MLP}(\kappa_S(k, i))}. \tag{8}$$

Note that $\kappa_{\mathrm{norm}}(k, i) \in [0, 1]$, so every teacher has a chance to be sampled. In addition, a teacher not sampled in the previous epoch does not mean that it will not be sampled in the next epoch, which allows our model to try a sufficient number of task combinations. At the convergence stage, the probability of some teachers will be close to 0 so that they hardly participate in knowledge distillation, which we call dynamic quit. It helps us select the most suitable teachers for each instance because the model can try enough task combinations and finally discard those useless tasks. To differentiably select $k$-th teacher for $i$-th instance according to the normalized sampling probability $\kappa_{\mathrm{norm}}(k, i)$, we use the reparameterization trick, which is commonly used in previous works (Huang et al., 2016a; Liu et al., 2021b). Specifically, we adopt Gumbel-Softmax sampling (Maddison et al., 2016), which samples $\kappa(k, i)$ from the Bernoulli distribution with probability $\kappa_{\mathrm{norm}}(k, i)$, defined as

$$\kappa(k, i) = \left\lfloor \frac{1}{1 + \exp^{-\left(\log \kappa_{\mathrm{norm}}(k, i) + G\right)/\tau}} + \frac{1}{2} \right\rfloor, \tag{9}$$

where $\tau = 1.0$ is the temperature of gumbel-softmax distribution, and $G \sim \mathrm{Gumbel}(0, 1)$ is a Gumbel random variable. Finally, we weigh only those selected teachers by re-softmax, as follows

$$\lambda(k, i) = \frac{\kappa(k, i) \exp(\zeta(k, i))}{\sum_{k=1}^{K} \kappa(k, i) \exp(\zeta(k, i))} \tag{10}$$

So far, our WAS framework, including weigh and select, has been built. Due to space constraints, pseudo-code and complexity analysis are placed in **Appendix B** and **Appendix C**.

## 5 EXPERIMENTAL EVALUATION

In this section, we evaluate WAS on 16 public graph datasets for both node-level and graph-level downstream tasks. To be specific, we would like to answer the following five questions: **Q1. (Improvement)** Does WAS achieve better performance than its corresponding teachers? **Q2. (Effectiveness)** How does WAS compare to other leading baselines? Can WAS's performance continue to grow as the task pool expands? **Q3. (Localization)** Can WAS learn customized task combinations for different instances? **Q4. (Decouple)** Whether the selecting results are decoupled from the weighing results? **Q5. (Integrity)** How does each component contribute to the performance of WAS?

**Datasets & Implementation Details.** For graph-level tasks, following Liu et al. (2022), we use 5-layers GIN (Xu et al., 2018) as our backbone. We randomly select 50k qualified molecules from GEOM (Axelrod & Gomez-Bombarelli, 2020) for the pre-training and then fine-tune on 8 graph datasets, including BACE, BBBP, ClinTox, SIDER, Tox21, Toxcast, MUV, and HIV. For node-level tasks, following Wu et al. (2022a), we use 2-layers GCN (Kipf & Welling, 2016) as our backbone. We conduct experiments on 8 real-world datasets widely used in the literature (Yang et al., 2016; Hassani & Khasahmadi, 2020a; Thakoor et al., 2021), i.e., Physics, CS, Photo, Computers, WikiCS, Citeseer, Cora, and ogbn-arxiv. A statistical overview of these datasets is placed in **Appendix D**. The implementation details and hyperparameter settings for each dataset are available in **Appendix E**.

**Task Pool.** There are a total of 12 classic graph pre-training tasks that make up our task pool. For graph-level tasks, we consider 7 pre-training tasks, including AttrMask (Hu et al., 2020b), ContextPred (Hu et al., 2020b) EdgePred (Hamilton et al., 2017), GPT-GNN (Hu et al., 2020c), GraphLoG (Xu et al., 2021), GraphCL (You et al., 2020a), and InfoGraph (Sun et al., 2019). For node-level tasks, we follow AutoSSL (Jin et al., 2021) to consider five tasks: DGI (Velickovic et al., 2019), CLU (You et al., 2020b), PAR (You et al., 2020b), PAIRSIM (Jin et al., 2020), and PAIRDIS (Peng et al., 2020). Detailed description of these tasks has been placed in **Appendix A**.

Table 2: Results of seven baseline teachers and WAS for molecular property prediction. For each dataset, we report the mean and standard deviation of ROC-AUC(%) with scaffold splitting. The best and second-best results are marked in **bold** and underlined, respectively. The arrows indicate whether the multi-task method has improved relative to the average performance of the seven teacher tasks.

| | BACE | BBBP | ClinTox | SIDER | Tox21 | Toxcast | MUV | HIV | Avg. Rank |
|---|---|---|---|---|---|---|---|---|---|
| AttrMask | 77.4±0.2 | 65.3±1.6 | 70.3±7.5 | 55.1±0.7 | 74.4±0.5 | 62.6±0.1 | 75.4±2.7 | 75.9±0.4 | 5.3 |
| ContextPred | 77.3±1.0 | 69.0±2.0 | 66.9±7.6 | 58.7±1.6 | 72.9±0.8 | 61.7±0.7 | 73.6±0.3 | 76.1±2.4 | 5.6 |
| EdgePred | 66.1±2.6 | 68.6±6.7 | 67.3±2.0 | 58.9±1.3 | 68.2±9.1 | 59.0±0.9 | 73.5±1.9 | 73.8±2.4 | 7.5 |
| GPT-GNN | 78.6±2.9 | 65.3±1.5 | 56.1±8.9 | 57.9±0.2 | 74.3±0.7 | 63.3±0.3 | 75.6±1.8 | 74.8±1.4 | 5.8 |
| GraphCL | 77.5±1.6 | 69.9±1.6 | 72.1±4.7 | 59.9±1.5 | 75.1±0.8 | 62.8±0.7 | 75.1±1.5 | 74.5±0.6 | 3.4 |
| GraphLoG | 78.1±1.0 | 66.4±2.8 | 64.1±3.4 | 59.5±2.4 | 73.9±1.4 | 62.3±0.6 | 73.5±1.0 | 75.5±0.5 | 5.5 |
| InfoGraph | 71.4±2.4 | 66.2±1.1 | 71.5±6.3 | 58.2±0.4 | 74.0±0.5 | 60.7±0.9 | 73.2±0.8 | 74.3±0.5 | 7.5 |
| Average-Weight | 75.3±1.2↑ | 66.6±0.2↓ | 62.6±4.4↓ | 56.9±1.5↓ | 67.8±0.8↓ | 62.2±1.8↑ | 74.2±0.7↑ | 74.8±0.3↓ | 7.5 |
| Random-Select | 77.2±1.7↑ | 67.2±0.5↓ | 58.1±1.5↓ | 59.2±1.7↑ | 74.3±0.6↓ | 59.3±0.7↓ | 71.6±1.4↓ | 76.1±1.2↑ | 6.3 |
| WAS | **80.7±0.5↑** | **70.5±1.0↑** | **73.1±0.5↑** | **60.4±0.4↑** | **75.4±0.7↑** | **63.7±1.5↑** | **76.3±1.0↑** | **78.4±1.9↑** | **1.0** |

### 5.1 PERFORMANCE COMPARISION

**A1: Performance comparision with Teachers.** To answer **Q1 (Improvement)**, we report the results for seven individual tasks and the model trained by WAS using the seven tasks as its teachers in Table 2. Besides, we add two additional combination methods, `Average-Weight` and `Random-Select` to better compare the performance. `Average-Weight` assigns equal weights to each teacher, and `Random-Select` randomly select teachers for each instance. From Table. 2, we can make the following observations: (1) There is no one optimal pre-training task that works for all datasets, which means we need to choose the most suitable pre-training tasks for each dataset separately. (2) Most of the results (9 out of 16) of `Average-Weight` and `Random-Select` are worse than the average performance, indicating that importance and compatibility issues are quite important and challenging. (3) WAS achieves the best performance on all the datasets, which means that it can handle the **compatibility** problem between tasks effectively in the vast majority of cases.

**A2: Performance comparison with other baselines.** For graph-level tasks (Table. 3), we compare WAS with other state-of-the-art graph pre-training baselines, including AD-GCL (Suresh et al., 2021),

Table 3: Comparison with leading graph pre-training baselines for the task of molecular property prediction, where the best and second are marked **bold** and underlined, respectively.

| | BACE | BBBP | ClinTox | SIDER | Tox21 | Toxcast | MUV | HIV | Avg. Rank |
|---|---|---|---|---|---|---|---|---|---|
| AD-GCL | 76.3±0.6 | 70.1±1.2 | 78.1±3.6 | 59.4±0.7 | 74.7±1.1 | 62.8±0.5 | 71.6±0.8 | 77.7±1.2 | 4.4 |
| R-GCL | 73.6±1.2 | 69.3±0.4 | 78.9±4.1 | 59.9±0.9 | 75.2±0.3 | 62.4±0.6 | 74.7±1.2 | 76.1±0.6 | 4.3 |
| G-Contextual | 76.3±1.9 | 64.0±1.5 | 67.6±2.9 | **61.1±2.5** | 74.9±0.2 | 61.7±0.1 | 74.2±1.4 | 75.3±1.4 | 5.6 |
| G-Motif | 71.5±0.9 | 70.3±1.5 | **81.4±2.2** | 60.1±1.7 | 73.1±0.4 | 61.8±0.6 | 74.5±0.6 | 76.0±1.3 | 5.4 |
| JOAO | 73.4±1.2 | 64.7±0.8 | 66.1±3.7 | 60.7±1.0 | 74.8±0.6 | 62.4±0.4 | 75.5±0.9 | 76.9±1.1 | 5.0 |
| GraphMAE | 75.0±0.7 | 66.3±0.7 | 65.3±3.9 | 57.1±0.7 | 68.8±0.5 | 61.5±0.5 | 71.8±0.3 | 73.5±0.8 | 8.3 |
| SimGRACE | 75.8±0.4 | 66.7±0.6 | 66.8±2.0 | 58.4±1.1 | 74.3±0.2 | 62.1±0.1 | 74.3±0.8 | 74.4±1.0 | 6.6 |
| GraphMVP | 80.3±1.4 | 67.9±1.0 | 76.4±1.1 | 60.2±1.6 | 74.6±0.3 | 63.5±0.2 | 75.5±1.6 | 76.2±0.8 | 3.8 |
| WAS | **80.7±0.5** | **70.5±1.0** | 73.1±0.5 | 60.4±0.4 | **75.4±0.7** | **63.7±1.5** | **76.3±1.0** | **78.4±1.9** | **1.8** |

Table 4: Performance comparison with baseline teachers and other methods for node classification. The *baseline* tasks form the task pool of the *multi-task* methods, while *single-task* represents other complex designed tasks. The Blue in line WAS indicates that it beats all multi-task methods. The results of AGSSL are from its original paper, and the others are reproduced on our platform.

| | | Cora | Citeseer | Pubmed | CS | Physics | Photo | Computers | ogbn-arxiv | Avg.Rank |
|---|---|---|---|---|---|---|---|---|---|---|
| baselines | CLU | 81.4±0.2 | 71.8±0.5 | 79.3±0.6 | 91.2±0.6 | 93.5±0.5 | 92.3±0.4 | 87.6±0.6 | 71.5±0.6 | 11.1 |
| | PAR | 82.4±0.3 | 71.6±0.6 | 79.6±0.4 | 91.3±0.5 | 93.1±0.8 | 92.4±0.5 | 86.7±0.8 | 71.3±0.4 | 11.4 |
| | PAIRSIM | 82.4±0.4 | 71.8±0.6 | 79.3±0.6 | 91.6±0.5 | 93.3±0.3 | 92.5±0.6 | 87.1±0.9 | 71.6±0.3 | 9.9 |
| | PAIRDIS | 81.9±0.6 | 72.2±0.5 | 79.6±0.5 | 91.4±0.5 | 94.0±0.6 | 92.1±0.5 | 86.5±0.7 | 71.4±0.3 | 10.6 |
| | DGI | 82.1±0.4 | 72.5±0.4 | 79.9±0.6 | 91.8±0.7 | 93.9±0.6 | 92.0±0.6 | 87.4±0.7 | 71.9±0.3 | 8.9 |
| single-task | GRLC | 82.8±0.1 | 73.2±0 1 | 80.8±0.2 | 89.9±0.2 | 93.2±0.5 | 91.9±0.6 | 88.1±0.4 | 71.0±0.7 | 9.4 |
| | CG3 | 83.4±0.4 | 73.6±0.2 | 81.1±0.1 | 91.7±0.4 | 93.6±0.6 | 92.1±0.5 | 87.8±0.9 | 71.4±0.9 | 7.1 |
| | GCA | 83.9±0.4 | 73.8±0.7 | 80.1±0.4 | **92.8±0.4** | 94.1±0.5 | 93.2±0.6 | 88.7±0.2 | OOM | 3.9 |
| | MVGRL | 83.4±0.4 | 72.7±0.6 | **81.6±0.8** | 91.7±0.2 | OOM | 93.1±0.4 | 87.9±0.2 | OOM | 6.3 |
| | GRACE | 80.1±0.6 | 72.9±0.7 | 79.7±0.6 | 91.5±0.4 | 94.5±0.4 | 92.4±0.3 | 88.5±0.5 | OOM | 8.3 |
| multi-task | ParetoGNN | 83.5±0.6 | 73.9±0.5 | 80.1±0.2 | 92.3±0.4 | 94.4±0.6 | 93.4±0.2 | 87.9±0.3 | 71.0±0.1 | 5.5 |
| | AutoSSL | 83.2±0.9 | 73.4±0.6 | 80.7±0.7 | 92.4±0.6 | 93.9±0.6 | 93.5±0.7 | 88.4±0.3 | 72.3±0.4 | 4.9 |
| | AGSSL | 84.2±0.3 | 73.6±0.6 | 80.6±0.3 | 92.4±0.5 | 94.8±0.3 | 93.3±0.4 | 88.7±0.4 | **72.5±0.3** | 3.1 |
| | WAS | **84.4±0.4** | **74.1±0.5** | 80.9±0.3 | 92.6±0.3 | **94.9±0.3** | **93.6±0.7** | **88.9±0.3** | 71.7±0.3 | **1.8** |

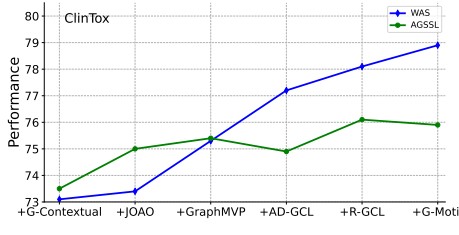

(a) Performance comparison on `ClinTox`

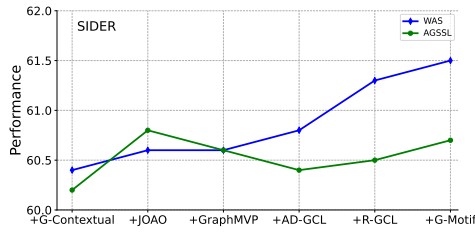

(b) Performance comparison on `SIDER`

Figure 4: Evaluation on whether the performance of WAS can improve as the task pool expands.

R-GCL (Li et al., 2022), GraphMVP (Liu et al., 2022), JOAO (You et al., 2021), GraphMAE (Hou et al., 2022), SimGRACE (Xia et al., 2022),G-Contextual and G-Motif (Rong et al., 2020). On 6 out of 8 datasets, WAS performs better than the other state-of-the-art methods, even though it combines only several simple but classical tasks. For node-level tasks (Table. 4), we compare WAS with other single-task methods, including GRLC (Peng et al., 2023), GCA (Zhu et al., 2020c), MVGRL (Hassani & Khasahmadi, 2020b), GRACE (Zhu et al., 2020a) and CG3 (Wan et al., 2020), and three multi-task methods, e.g., ParetoGNN (Ju et al., 2022), AutoSSL (Jin et al., 2021) and AGSSL (Wu et al., 2022a). Here we would like to emphasize that the most important contribution of WAS is the handling of the compatibility, i.e., the continuous performance improvement that WAS can achieve as the number of tasks increases. To demonstrate this, we extend AGSSL to the graph-level tasks and compare it with WAS. As shown in Fig. 4, the performance of WAS improves consistently as the number of tasks gradually increases, but the performance of AGSSL hardly benefits from more tasks.

## 5.2 EVALUATION ON INSTANCE-LEVEL RESULTS

**A3: Customized combinations for different instances.** To show the different selecting results for different instances, we chose 4 molecular graphs from `Bace` and visualized the probability of being selected over them for each task in Fig. 5. We can see that the selecting module can select customized combinations for each instance even if we do not pre-specify the number of selected teachers.

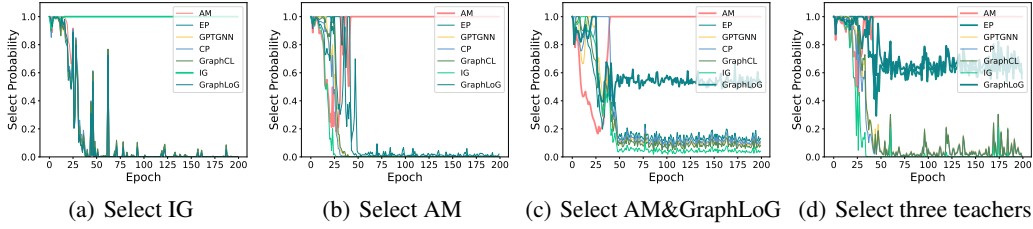

(a) Select IG   (b) Select AM   (c) Select AM&GraphLoG   (d) Select three teachers

Figure 5: Probability of being selected for different teachers (tasks) on different instances from `Bace`
.

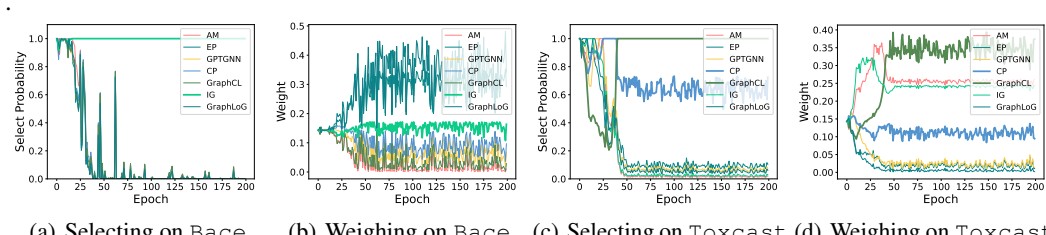

(a) Selecting on `Bace`   (b) Weighing on `Bace`   (c) Selecting on `Toxcast`   (d) Weighing on `Toxcast`

Figure 6: Comparison of the selecting probabilities and importance weights for different instances.
(a)(b) selecting IG (rangking third importance) as the teacher for an instance from `Bace`. (c)(d)
selecting GraphCL and CP (ranking fourth importance) as teachers for an instance from `Toxcast`.

**A4: Evolution process of selecting and weighing.** Here, we select 2 molecular graphs from `Bace`
and `Toxcast` to visualize the results of selecting and weighing. It can be seen from Fig. 6 that the
selected teacher is not necessarily the teacher with the highest weight, which confirms our earlier
statement that two modules need to be decoupled to deal with two issues of the **importance** and
**compatibility** separately, and also demonstrates the effectiveness of our decoupled siamese networks.

## 5.3 ABLATION STUDY

**A5: Ablation study on each component.** The results in Table. 5 provide
some constructive insights: (1) The
performance of *Importance-based* selecting can be even lower than that
of *Random* selecting. This confirms
our claim that the **compatibility** issue needs to be considered separately
from the **importance** issue. (2) Removing the projection head (implemented as MLP) would be detrimental
to performance, as it would increase
the risk of coupling the result of se-

Table 5: Mean and standard deviation of ROC-AUC(%). *Random* means randomly selecting teachers. *All* means selecting
all teachers. *Importance-based* means selecting the top-3
important teachers. All of them use learned weights.

|  | BACE | HIV | Tox21 | BBBP |
|---|---|---|---|---|
| Random | 77.2±1.7 | 76.1±1.2 | 74.3±0.6 | 67.2±0.5 |
| All | 75.2±1.2 | 77.8±0.7 | 73.6±0.9 | 65.9±0.4 |
| Importance-based | 75.4±0.7 | 73.5±1.2 | 74.0±0.4 | 64.4±0.9 |
| WAS w/o MLP | 77.4±1.1 | 75.4±0.9 | 73.9±0.8 | 69.6±1.2 |
| WAS w/o re-weighing | 65.1±3.7 | 63.3±2.4 | 64.1±3.0 | 60.7±5.1 |
| WAS | **80.7±0.5** | **78.4±1.9** | **75.4±0.7** | **70.5±1.0** |

lecting with weighing, which highlights the role of the projection head in the selection module. (3)
WAS outperforms all other selecting strategies, which indicates that a proper selecting strategy can
greatly improve performance. (4) Re-weighing after selection is very important. If we remove the
re-weighing, the final sum of the teacher weights will not equal 1, which may result in the number of
selected teachers having a significant impact on performance. Due to space limitations, some more
related experiments on hyperparameter sensitivity and teacher selection are placed in **Appendix F**.

## 6 CONCLUSION

With hundreds of graph pre-training tasks proposed, combining the information drawn by multiple pre-
training tasks has become a hot research topic. In this paper, we identify two important collaborative
processes for this combination operation: *weigh* and *select*; we provide extensive experiments to
explain why the collaboration of the two is important, how it differs from the *weighing*-only based
methods and why do the two need to be decoupled. Furthermore, we propose a novel decoupled
framework WAS that is capable of customizing a selecting & weighing strategy for each downstream
instance separately. Extensive experiments on 16 graph datasets show that WAS can not only achieve
comparable performance to other leading counterparts by selecting and weighing several simple but
classical tasks, but also achieve consistent performance improvements as the task pool expands.

## 7  ACKNOWLEDGMENTS

This work was supported by National Key R&D Program of China (No. 2022ZD0115100), National Natural Science Foundation of China Project (No. U21A20427), and Project (No. WU2022A009) from the Center of Synthetic Biology and Integrated Bioengineering of Westlake University.

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

# Appendix

## A. DETAILS ON PRE-TRAINING TASKS

Here, we provide a high-level comparison among existing graph pre-training tasks in Table. A1.

**Generative tasks:** The generative tasks focus on the intra-data information embedded in the graph and aim to reconstruct the important structures or features for each graph. By doing so, it learns a better representation that encodes the key components of the data.

**Contrastive tasks:** The contrastive tasks handle inter-data information by first applying transformations to construct different views for each graph. Each view contains information at a different granularity. The training objective is to align the representations of views from the same data (positive pairs) and pull apart the representations of views from different data (negative pairs).

**Predictive tasks:** The predictive methods generally self-generate labels by some simple statistical analysis or expert knowledge and then perform prediction-style tasks based on self-generated labels.

Table A1: Comparison between existing pre-training tasks.

| Pre-training Tasks | Generative | Contrastive | Predictive |
|---|:---:|:---:|:---:|
| EdgePred (Hamilton et al., 2017) | ✓ | - | - |
| AttrMask (Hu et al., 2019) | ✓ | - | - |
| GPT-GNN (Hu et al., 2020c) | ✓ | - | - |
| GraphMVP-G (Liu et al., 2022) | ✓ | - | - |
| DGI (Velickovic et al., 2019) | - | ✓ | - |
| InfoGraph (Sun et al., 2019) | - | ✓ | - |
| ContextPred (Hu et al., 2019) | - | ✓ | - |
| G-Contextual (Rong et al., 2020) | - | ✓ | - |
| GraphCL (You et al., 2020a) | - | ✓ | - |
| GraphLoG (Xu et al., 2021) | - | ✓ | - |
| AD-GCL (Suresh et al., 2021) | - | ✓ | - |
| JOAO (You et al., 2021) | - | ✓ | - |
| SimGRACE (Xia et al., 2022) | - | ✓ | - |
| R-GCL (Li et al., 2022) | - | ✓ | - |
| GraphMVP-C (Liu et al., 2022) | - | ✓ | - |
| GraphMAE (Hou et al., 2022) | - | - | ✓ |
| PAIRSIM (Jin et al., 2020) | - | - | ✓ |
| PAIRDIS (Peng et al., 2020) | - | - | ✓ |
| CLU (You et al., 2020b) | - | - | ✓ |
| PAR (You et al., 2020b) | - | - | ✓ |
| G-Motif (Rong et al., 2020) | - | - | ✓ |

**Baseline.** For graph-level classification tasks, we take seven classic graph pre-training tasks to make up our task pool. (1) AttrMask (Hu et al., 2020b), which learns the regularities of node/edge attributes. (2) ContextPred (Hu et al., 2020b), which explores graph structures by predicting the contexts. (3) EdgePred (Hamilton et al., 2017), which predicts the connectivity of node pairs. (4) GPTGNN (Hu et al., 2020c), which introduces an attributed graph generation task to pre-train GNNs. (5) GraphLoG (Xu et al., 2021), which introduces a hierarchical prototype to capture the global semantic clusters. (6) GraphCL (You et al., 2020a), which constructs specific contrastive views of graph data. (7) InfoGraph (Sun et al., 2019), which maximizes the mutual information between the representations of the graph and substructures. For node-level classification tasks, we follow AutoSSL (Jin et al., 2021) to adopt five classic tasks. (1) DGI (Velickovic et al., 2019), which maximizes the mutual information between graph representation and node representation. (2) CLU (You et al., 2020b), which predicts pseudo-labels from $K$-means clustering on node features. (3) PAR (You et al., 2020b), which predicts pseudo-labels from Metis graph partition (Karypis & Kumar, 1998). (4) PAIRSIM (Jin et al., 2020), which predicts pairwise feature similarity between nodes. (5) PAIRDIS (Peng et al., 2020), which predicts the shortest path length between nodes.

## B. PSEUDO CODE

The pseudo-code of WAS is summarized in Algorithm 1 (node-level) and Algorithm 2 (graph-level).

---

**Algorithm 1** Algorithm of the WAS framework for node-level tasks

---

**Input:** $K$ Pre-training Tasks $T_1, T_2, \cdots, T_K$, Graph $\mathcal{G} = (\mathbf{A}, \mathbf{X})$ with $N$ nodes, and Number of Epochs: $Iteration$.
**Output:** GNN Encoder $h_\theta(\cdot)$ and Projection Head $g_\omega(\cdot)$.
Use $K$ pre-training tasks to pre-train GNN encoder to get teacher models' parameters $\{\theta_k^*, \omega_k^*\}_{k=1}^K$.
**for** $iter = 1, 2, \cdots, Iteration$ **do**
    **for** $i = 1, 2, \cdots, N$ **do**
        Save the output $\left\{\mathcal{P}(k,i) = g_{\omega_k^*}(h_{\theta_k^*}(\mathbf{x}_i))\right\}_{k=1}^K$ on $\mathcal{G}$ from the pre-trained teachers and then freeze them.
        Input $\mathcal{P}(k,i)$ into Weighing and Selecting module to obtain the weights $\lambda_{init}(k,i)$ and selecting results $\kappa(k,i)$ of each teacher.
        Weigh those selected teachers with weights $\lambda(k,i) = \frac{\kappa(k,i)exp(\lambda_{init}(k,i))}{\sum_{k=1}^K \kappa(k,i)exp(\lambda_{init}(k,i))}$.
        Integrate the knowledge of different teachers by $\mathcal{P}^T(\mathbf{x}_i) = \sum_{k=1}^K \kappa(k,i)\lambda(k,i)\mathcal{P}(k,i)$.
        Distill the integrated knowledge to the student model by Eq. (5).
    **end for**
**end for**
**Return:** Student's Enocder $h_\theta(\cdot)$ and Prediction Head $g_\omega(\cdot)$.

---

**Algorithm 2** Algorithm of the WAS framework for graph-level tasks

---

**Input:** $K$ Pre-training Tasks $T_1, T_2, \cdots, T_K$, a set of graph $G = \{\mathcal{G}_1, \mathcal{G}_2, \cdots, \mathcal{G}_M\}$, and Number of Epochs: $Iteration$.
**Output:** GNN Enocder $h_\theta(\cdot)$ and Projection Head $g_\omega(\cdot)$.
Use $K$ pre-training tasks to pre-train GNN encoder to get teacher models' parameters $\{\theta_k^*, \omega_k^*\}_{k=1}^K$.
**for** $iter = 1, 2, \cdots, Iteration$ **do**
    **for** $i = 1, 2, \cdots, M$ **do**
        Save the output $\left\{\mathcal{P}(k,i) = g_{\omega_k^*}(h_{\theta_k^*}(\mathcal{G}_i))\right\}_{k=1}^K$ from the pre-trained teachers and then freeze them.
        Input $\mathcal{P}(k,i)$ into Weighing and Selecting module to obtain the weights $\lambda_{init}(k,i)$ and selecting results $\kappa(k,i)$ of each teacher.
        Weigh those selected teachers with weights $\lambda(k,i) = \frac{\kappa(k,i)exp(\lambda_{init}(k,i))}{\sum_{k=1}^K \kappa(k,i)exp(\lambda_{init}(k,i))}$.
        Integrate the knowledge of different teachers by $\mathcal{P}^T(\mathcal{G}_i) = \sum_{k=1}^K \kappa(k,i)\lambda(k,i)\mathcal{P}(k,i)$.
        Distill the integrated knowledge to the student model by Eq. (5).
    **end for**
**end for**
**Return:** Student's Enocder $h_\theta(\cdot)$ and Prediction Head $g_\omega(\cdot)$.

---

## C. COMPLEXITY ANALYSIS

(1) **[Few additional parameters]** Despite the additional parameters introduced, WAS achieves a similar computational efficiency as previous works, because the training with multiple tasks is more hard to optimize than the training with one single task. The two processes, weigh and select, actually use only $2KT(C + 1) + T^2$ more parameters ($T$ is teacher numbers, $K$ is the dimension of the label distribution, and $C$ is the dimension of latent space), which is acceptable, as $T$ is usually small.

(2) **[Few additional memory]** Since we adopt a multi-teacher architecture, we can not only train multiple teachers in parallel, but also save only their outputs instead of the full model parameters.

(3) **[Few additional time consumption]** In Table. A2, we demonstrate the training time (in the pre-training and fine-tuning setting) of a single method (Clu and DGI) and WAS (train in parallel) here. Specifically, to illustrate that the two processes, weigh and select, will not cause an unacceptable time consumption, we additionally compare the case without them.

(4) **[Few additional Time complexity]** The time complexity of WAS is based on: 1. Training teachers: $O(T(||A||F + NdF))$, $N$ is the number of instances, $||A||$ is the number of non-zero values in the adjacency matrix, and $d$ and $F$ are the dimensions of input and hidden spaces. 2. Knowledge

integrating: $O((N + 1)FT)$. The overall complexity is still determined by the first term. Moreover, we can reduce the complexity of training teachers to $O(||A||F + NdF)$ by parallel training.

Table A2: Training time of a single method (Clu and DGI) and WAS (train in parallel).

|  | Cora | Citeseer | Pumbed | Photo |
|---|---|---|---|---|
| Clu | 6.75s | 9.57s | 28.39s | 11.95s |
| DGI | 9.18s | 9.26s | 26.23s | 12.06s |
| WAS (w/o select&weigh) | 11.73s | 12.19s | 32.17s | 15.79s |
| WAS (w/o select) | 12.04s | 12.61s | 32.82s | 16.23s |
| WAS | 12.97s | 12.76s | 34.07s | 16.78s |

## D. DATASET STATISTICS

In this section, we summarize 16 datasets used in this paper.

**Dataset about Pharmacology**: The Blood-Brain Barrier Penetration (BBBP) (Martins et al., 2012) dataset measures the penetration properties of a molecule into the central nervous system. The Side Effect Resource (SIDER) (Kuhn et al., 2016) dataset represents the adverse drug reactions. Tox21 (Huang et al., 2016b), ToxCast (Richard et al., 2016), and ClinTox (Gayvert et al., 2016) are related to the toxicity of molecular compounds.

**Dataset about Biophysics**: Maximum Unbiased Validation (MUV) (Rohrer & Baumann, 2009) using refined nearest neighbor analysis to construct based on PubChem (Kim et al., 2016). HIV (Zaharevitz, 2015) aims at predicting inhibit HIV replication. BACE, which is gathered in MoleculeNet (Wu et al., 2018), measures the binding results for inhibitors of $\beta$-secretase 1. Following the setting of (Liu et al., 2022), we use scaffold splitting to split datasets into train/val/test sets by ratios of 0.7/0.2/0.1.

Table A3: Statistical information of the graph-level datasets.

| Dataset | BACE | BBBP | ClinTox | Sider | Tox21 | ToxCast | MUV | HIV |
|---|---|---|---|---|---|---|---|---|
| # Tasks | 1 | 1 | 2 | 27 | 12 | 617 | 17 | 1 |
| # Molecules | 1,513 | 2,039 | 1,478 | 1,427 | 7,831 | 8,576 | 93,087 | 41,127 |

**Dataset about Citation network**: Cora (Sen et al., 2008), Citeseer (Giles et al., 1998), ogbn-arxiv (Hu et al., 2020a) and Pubmed (McCallum et al., 2000) consists of papers from different fields and each node is a scientific paper.

**Dataset about Co-authorship network**: Coauthor-CS (CS) and Coauthor-Physics (Physics) (Shchur et al., 2018) are Co-authorship network datasets consisting of undirected graphs, where nodes represent authors and are connected by edges if they are co-authors. Different authors have different research areas.

**Dataset about Co-purchase network**: Amazon-Photo (Photo) and Amazon-Computers (Computers) (Shchur et al., 2018) are datasets consisting of an undirected graph, where the nodes represent goods and the edges represent two goods that are often purchased at the same time. When we evaluate the performance of node classification, we need to use the training and testing data. For Cora, Citeseer, and Pubmed, we follow the data splitting strategy in (Kipf & Welling, 2016). For CS, Physics, Photo, and Computers, we follow (Zhang et al., 2021; Luo et al., 2021) to randomly split the data into train/val/test sets, and each random seed corresponds to a different splitting. For ogbn-arxiv, we use the public data splits provided by the authors (Hu et al., 2020a).

## E. DETAILS ON EXPERIMENTAL SETUP

**Experimental settings.** For graph-level tasks, we follow the setting of GraphMVP (Liu et al., 2022). We pre-trained all methods on the same dataset based on GEOM (Axelrod & Gomez-Bombarelli, 2020) using the GIN (Xu et al., 2018) as the backbone model, and then we fine-tune them on 8

Table A4: Statistical information of the node-level datasets.

| Dataset | Cora | Citeseer | Pubmed | Photo | CS | Physics | Computers | ogbn-arxiv |
|---|---|---|---|---|---|---|---|---|
| # Nodes | 2,708 | 3,327 | 19,717 | 7,650 | 18,333 | 34,493 | 13,752 | 169,343 |
| # Edges | 5,278 | 4,614 | 44,324 | 119,081 | 81,894 | 247,962 | 245,861 | 1,166,243 |
| # Features | 1,433 | 3,703 | 500 | 745 | 6,805 | 8,415 | 767 | 128 |
| # Classes | 7 | 6 | 3 | 8 | 15 | 5 | 10 | 40 |

classical molecular property prediction datasets. For a fair comparison, the model with the highest validation accuracy is selected for testing. For node-level tasks, we follow the settings in AGSSL (Wu et al., 2022a) exactly (search space of hyperparameter, model architecture, experiment setting, etc.) for a fair comparison. For pre-training, we pre-train the GNN encoder (using GCN (Kipf & Welling, 2016)) with pre-training tasks and for fine-tuning, we use the pre-trained GNN encoder with a projection head under the supervision of a specific downstream task. We conduct our experiments on NVIDIA Tesla V100 GPU and we use Intel(R) Xeon(R) Gold 6240R @ 2.40GHz CPU.

**Hyperparameter Settings.** The following hyperparameters are set the same for all molecular datasets: Adam optimizer with weight decay $w = 0$; Epoch $E = 250$ ($E = 500$ for `ClinTox` and `Toxcast`). The other hyperparameters settings (including loss weight $\alpha$, temperature $\tau$, momentum rate $m$, and learning rate $lr$) are included in Table. A5. The following hyperparameters are set the same for node classification tasks: Adam optimizer with weight decay $w = 5e - 4$; Epoch $E = 500$. The other hyperparameter settings (including hidden dimension $F$) are included in Table. A6.

Table A5: Hyperparameters of graph-level tasks.

|  | BACE | BBBP | ClinTox | SIDER | Tox21 | Toxcast | MUV | HIV |
|---|---|---|---|---|---|---|---|---|
| $\alpha$ | 3.6 | 3.3 | 0.4 | 4.2 | 4.1 | 2.7 | 1.6 | 4.2 |
| $\tau$ | 3.5 | 1.1 | 4.9 | 1.3 | 1.6 | 1.5 | 1.5 | 1.5 |
| $m$ | 0.9 | 0.9 | 0.9 | 0.5 | 0.1 | 0.5 | 0.5 | 0.3 |
| $lr$ | 0.01 | 0.01 | 0.001 | 0.01 | 0.01 | 0.001 | 0.01 | 0.01 |

Table A6: Hyperparameters of node-level tasks.

|  | Cora | Citeseer | Pubmed | CS | Physics | Photo | Computers | ogbn-arxiv |
|---|---|---|---|---|---|---|---|---|
| $F$ | 128 | 256 | 512 | 64 | 256 | 64 | 128 | 512 |
| $\alpha$ | 1 | 20 | 10 | 5 | 10 | 0.5 | 0.1 | 1 |
| $\tau$ | 1.2 | 1 | 1 | 1 | 1.5 | 1.2 | 2 | 1.2 |
| $m$ | 0.3 | 0.5 | 0.5 | 0.3 | 0.5 | 0.7 | 0.7 | 0.7 |
| $lr$ | 0.01 | 0.01 | 0.01 | 0.01 | 0.01 | 0.01 | 0.01 | 0.001 |

## F. MORE RELATED EXPERIMENTS

### EVALUATION ON HYPERPARAMETER SENSITIVITY

The hyperparameter sensitivity on the loss weight $\alpha$ is provided in Fig. A1, from which we can make two observations that (1) As the loss weight $\alpha$ increases, the model performance first improves and then decreases, which suggests that knowledge transferring helps to improve performance, but too large an $\alpha$ can cause the loss directly associated with the downstream task to be masked, causing performance degradation. (2) Larger $\alpha$ leads to smaller variance, probably because combining the knowledge of multiple pre-training tasks can effectively improve the stability of the student model.

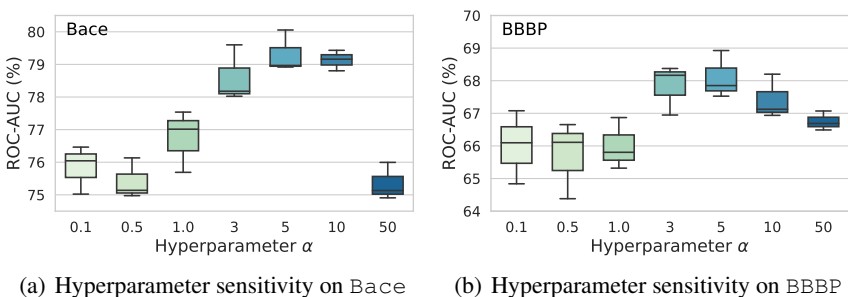

(a) Hyperparameter sensitivity on `Bace`  (b) Hyperparameter sensitivity on `BBBP`

Figure A1: Hyperparameter sensitivity analysis on the loss weight $\alpha$.

### EVALUATION ON TEACHER SELECTION

In Table. A7, we count the average number of selected teachers, and then count the ratio of selecting Top-1/Top-[number] important teacher. From the results reported in Table. A7, we can draw the following conclusions: (1) teachers with the highest weights have a higher probability of being selected, and (2) the selection module does not necessarily always select the higher-weight teachers.

Table A7: Average number of teachers being selected, and the ratios of Top-1/Top-[number] important teachers. The selection module does not necessarily always select the higher-importance teachers.

|  | Cora | Citeseer | BACE | BBBP |
| --- | --- | --- | --- | --- |
| average number of selected teachers | 2.7 | 3.9 | 2.9 | 3.1 |
| ratios of selecting Top-1 important teacher (%) | 0.63 | 0.71 | 0.68 | 0.55 |
| ratios of selecting Top-[number] important teacher (%) | 0.26 | 0.34 | 0.41 | 0.10 |

