# OpenReview forum: "Decoupling Weighing and Selecting for Integrating Multiple Graph Pre-training Tasks"
_ICLR.cc/2024/Conference — ICLR 2024 poster_

### Official Review · Reviewer_66Ho · 2023-10-28

**Soundness:** 2 fair
**Presentation:** 3 good
**Contribution:** 2 fair
**Rating:** 5
**Confidence:** 3

**Summary:**

This paper focus on multi-tasking graph pre-training and proposes a weighting and selecting network to model the compatibility and importance of tasks. The proposed WAS consists of knowledge extraction and transfer step, and it is powered by the decoupled siamese networks to assign weights to each teacher and do selection. Experiments are conducted on different benchmark datasets to showcase its effectiveness.

**Strengths:**

1. The paper is well-written and easy to follow.

2. The evaluation is conducted on both graph-level and instance-level tasks.

3. The idea on solving the compatibility issue of multiple tasks is new and insightful.

**Weaknesses:**

1. The motivation of focusing on graph-structured data is unclear.

2. Compared to the baseline methods for multi-task learning on instance-level, the improvement from WAS is marginal. On the largest graphs in the experiment (ogbn-arxiv), it achieves worse performance compared to baseline method.

3. Besides the empirical results, the theoretical analysis is still needed to answer why both selecting and weighing is needed,  and why there is compatibility issue (i.e., why some tasks shouldn't be selected).

**Questions:**

1. The proposed method does not treat graph-structured data specially, nor is it tailored for graph modeling. Why is it positioned for pre-training on graphs, instead of for general pre-training?

2. Any in-depth understanding of the relationship for the selected tasks? Are they independent?

---

> ### Author Response · Authors · 2023-11-17
> **Response to Reviewer 66Ho (1/2)**
>
> Dear Reviewer 66Ho:
>
>
> Thank you for your valuable feedback and constructive suggestions. We sincerely appreciate your acknowledgment of effectiveness and writing quality. In the revised manuscript, we have highlighted existing content (which may have been omitted, but which we would like to bring to your attention) in red and revised or new content in blue. Our detailed response to your concerns is listed as follows:
>
> ------
>
> **W1.&Q1.**  The motivation of focusing on graph-structured data is unclear. The proposed method does not treat graph-structured data specially, nor is it tailored for graph modeling. Why is it positioned for pre-training on graphs, instead of for general pre-training?
>
> **A1.**
>
> The reason for your confusion may be because we have not added too many graph-specific priors to our method. This is due to the diversity of downstream tasks in the graph domain, where different downstream tasks contain different instance levels, e.g., graph classification and node classification. This leads to the fact that imposing too many restrictions on a certain method can make it difficult to handle diverse downstream tasks, since the tasks employed in the method already provide domain-specific a priori (e.g., node masking, edge prediction, etc.). For example, the previous work AutoSSL\[1], used graph homophily as a priori knowledge, which caused it to only be able to handle node-level tasks and not graph-level tasks. We note that this series of subsequent work (ParetoGNN\[2], AGSSL\[3]) follows the idea of not imposing a priori on methods. Our starting point is common, i.e., imposing too many a priori on combinatorial methods can limit the scenarios in which the methods can be used.
>
> As with previous tasks, the reason we are a framework positioned on graph-structured data is that the phenomena we observe (sharp performance degradation due to conflicts among graph pre-training tasks, etc.) are graph-based. In addition, our method does show potential for extension to other domains, but we do not believe that this should be one of the drawbacks, and we will consider extensions of our method as meaningful follow-on work.
>
>
>
> \[1]Wei Jin, Xiaorui Liu, Xiangyu Zhao, Yao Ma, Neil Shah and Jiliang Tang, Automated Self-Supervised Learning for Graphs, ICLR2022
>
> \[2] Mingxuan Ju, Tong Zhao, Qianlong Wen, Wenhao Yu, Neil Shah, Yanfang Ye and Chuxu Zhang, Multi-task Self-supervised Graph Neural Networks Enable Stronger Task Generalization, ICLR 2023
>
> \[3] Lirong Wu, Yufei Huang, Haitao Lin, Zicheng Liu, Tianyu Fan and Stan Z. Li. Automated graph self-supervised learning via multi-teacher knowledge distillation. arxiv2210.02099
>
>
>
>
> ------
>
>
>
> **W2.** Compared to the baseline methods for multi-task learning on instance-level, the improvement from WAS is marginal. On the largest graphs in the experiment (ogbn-arxiv), it achieves worse performance compared to baseline method.
>
> **A2.**
>
> We kindly point out that "the improvement from WAS is marginal" might be somewhat inappropriate.
>
> Firstly, the major contribution of our method lies in its ability to **continually achieve performance gains from an expanding task pool**, a capability not possessed by previous methods. Besides this primary contribution, our method outperforms previous multi-task learning methods in terms of performance on seven node-level datasets, even when the task pool is kept the same. Additionally, our method is capable of handling **graph-level tasks**.

---

> ### Author Response · Authors · 2023-11-17
> **Response to Reviewer 66Ho (2/2)**
>
> **W3.&Q2.** Besides the empirical results, the theoretical analysis is still needed to answer why both selecting and weighing is needed, and why there is compatibility issue (i.e., why some tasks shouldn't be selected). Any in-depth understanding of the relationship for the selected tasks? Are they independent?
>
> **A3.**
>
> We sincerely appreciate your suggestions for enhancing the clarity of our manuscript, which will make it stronger!  We have provided relevant analysis in Fig.1 of the Introduction, where Fig.1b illustrates the importance of a reasonable weighting scheme, and Fig.1c demonstrates the compatibility issues between different tasks. You may want us to include more detailed explanations regarding these issues.
>
> We will address your concerns in three steps: 1. Are different tasks relatively independent? 2. Why do compatibility issues arise? 3. Theoretically speaking, what are the advantages of incorporating a selection mechanism as opposed to simply using a weighting approach? For the figures and proofs mentioned in the following content, please refer to the Appendix G.
>
> - **[Are tasks independent? Different teachers acquired different knowledge during pre-training, and this knowledge remains different after migration to downstream tasks]** In Fig. A2, we analyze the knowledge acquired by different teachers. Different pre-training methods, e.g., GraphCL performs contrastive learning between instances, while AM masks node attributes on a single instance, are able to acquire different knowledge. Fig. A2 visualizes the representations obtained by different teachers after training on the same pre-training dataset by T-sne. It can be seen that they are distributed in different spaces, which means that **they do not acquire the same knowledge.** We can conclude that **the knowledge they acquire is essentially independent**.
> - **[Why do compatibility issues arise? Different instances may require different knowledge.]** Then, for downstream tasks, what differences do different teachers' acquired knowledge make? **Fig. A3** visualizes the graphs from the test set of BACE, where blue dots represent molecular graphs that can be correctly predicted for all seven teachers, while red dots represent graphs that can only be correctly predicted by some of the seven teachers. The red dots make up the majority, which means that **different teachers will be able to predict different graphs.** Therefore, for a given instance, if  a teacher whose knowledge is not suitable for them is chosen, this may cause conflicts and could be worse than not having that teacher at all. This is the compatibility issue.
> - **[What are the advantages of incorporating a selecting mechanism? Having only the weighting module can lead to an inability to improve performance.]** In the Appendix G, we provide a theoretical analysis of the performance without the selecting module (only the weighting module). Please refer to the Appendix G. Through this analysis, we can conclude that the selection module is a crucial factor for the model to achieve continuous performance improvement from an ever-expanding task pool.
>
> More detailed content (figures and theoretical analysis) has been added to the **Appendix G**. Please refer to **Appendix G** for further information.
>
> ---
>
> In light of these responses, we hope we have addressed your concerns, and hope you will consider increasing your score. If we have left any notable points of concern unaddressed, please share and we will attend to these points.

---

> ### Author Response · Authors · 2023-11-21
> **Looking forward to hearing from you**
>
> Dear Reviewer 66Ho,
>
> We sincerely thank you for taking the time to review our manuscript and providing valuable suggestions.
>
> Unlike previous years, there will be no second stage of author-reviewer discussions this year, and a recommendation needs to be provided by November 22, 2023. Considering that the author-reviewer discussion phase is nearing the end, we would like to be able to confirm whether our responses have addressed your concerns.
>
> We have provided detailed replies to your concerns a few days ago, and we hope we have satisfactorily addressed your concerns. If so, could you please consider increasing your rating? If you still need any clarification or have any other concerns, please feel free to contact us and we are happy to continue communicating with you.
>
> Best,
>
> Authors

---

> ### Comment · Reviewer_66Ho · 2023-11-22
> **Thanks for your responses**
>
> Having thoroughly reviewed the responses and comments from other reviewers, I've decided to keep my original score.

---

### Official Review · Reviewer_pHBL · 2023-11-01

**Soundness:** 3 good
**Presentation:** 3 good
**Contribution:** 3 good
**Rating:** 6
**Confidence:** 3

**Summary:**

In this paper, the authors have studied how to effectively integrate multiple graph pre-training tasks. They have identified two important collaborative processes, i.e., selecting and weighing.  They propose a new instance-level framework for integrating multiple graph pre-training tasks, named WAS (Weigh And Select), where the weighing and selecting processes are combined by decoupled siamese networks. Extensive experiments on 16 graph datasets have been performed to demonstrate the effectiveness of the proposed method.

**Strengths:**

1. In this paper, the authors introduce a new framework, namely WAS, for task selecting and importance weighing to integrate multiple graph pre-training tasks.

2. The authors show the limitations of existing weighing-only schemes and demonstrate the importance of task selecting process.

3. The authors have performed extensive experiments to demonstrate the effectiveness of the proposed method.

4. This paper is clearly written and easy to follow.

**Weaknesses:**

1. In Table 1, 2, and 3, the authors do not compare the proposed WAS framework with existing frameworks that only focus on weighing (i.e., AutoSSL, ParetoGNN, AUX-TS, and AGSSL). However, in Table 4, WAS is compared with these methods. This makes the experimental settings not consistent.

2. The complexity of the proposed has not been studied. Although the proposed method can achieve some performance improvement, it may take more training/inference time, due to combining multiple pre-training tasks. The authors need to have some analysis about the complexity of the proposed method.

3. The authors also need to perform experiments to analyse how many tasks are selected for each instance on average.

**Questions:**

1. On average, how many tasks are usually selected for each instance?

2. From Figure 6(d), we can observe that the following 4 tasks have the largest weights, i.e, EP, AM, IG, and CP. How about we only use these 4 tasks and learning the their weights for prediction?

---

> ### Author Response · Authors · 2023-11-17
> **Response to Reviewer pHBL**
>
> Dear Reviewer pHBL:
>
>
> Thank you for your valuable feedback and constructive suggestions. We sincerely appreciate your acknowledgment of effectiveness and writing quality. In the revised manuscript, we have highlighted existing content (which may have been omitted, but which we would like to bring to your attention) in red and revised or new content in blue.
>
> Our detailed response to your concerns is listed as follows:
>
> ------
>
> **W1.** In Table 1, 2, and 3, the authors do not compare the proposed WAS framework with existing frameworks that only focus on weighing (i.e., AutoSSL, ParetoGNN, AUX-TS, and AGSSL). However, in Table 4, WAS is compared with these methods. This makes the experimental settings not consistent.
>
> **A1.**
>
> We kindly point out that, in Table 1, we have compared these works.
>
> In Table 2 and 3, we did not compare these works because these methods are primarily focused on **node-level tasks**, rather than **graph-level tasks**. Methods like AutoSSL struggle with the **graph-level tasks** that we addressed in Table 2 and 3, making a comparison unfeasible. Since Table 4 concerns **node-level tasks**, we were able to compare our method with these approaches.
>
>
>
> ------
>
>
> **W2.** The complexity of the proposed has not been studied. Although the proposed method can achieve some performance improvement, it may take more training/inference time, due to combining multiple pre-training tasks. The authors need to have some analysis about the complexity of the proposed method.
>
> **A2.**
>
> We have provided an analysis of additional parameters, extra storage space, additional time consumption, and extra time complexity in **Appendix C (complexity analysis)**, and mentioned this at the end of Sec. 4.2.3 (Line 267).
>
>
>
> ------
>
>
> **W3.&Q1.** The authors also need to perform experiments to analyze how many tasks are selected for each instance on average. On average, how many tasks are usually selected for each instance?
>
> **A3.**
>
> We sincerely appreciate your suggestions for enhancing the clarity of our manuscript, which will make it stronger! In the following Table, we count the average number of selected teachers *number*, and then count the *ratio of selecting Top-1/Top-[number] important teacher*.
>
> From the table, we can draw the following conclusions:
>
> 1. teachers with the highest weights have a higher probability of being selected
> 2. the selection module does not necessarily always select the higher weight teachers
>
> |                                                     | Cora | Citeseer | BACE | BBBP |
> | :-------------------------------------------------- | :--- | :------- | :--- | :--- |
> | *number*                                            | 2.7  | 3.9      | 2.9  | 3.1  |
> | *ratio of selecting Top-1 important teacher*        | 0.63 | 0.71     | 0.68 | 0.55 |
> | *ratio of selecting Top-[number] important teacher* | 0.26 | 0.34     | 0.41 | 0.10 |
>
> This table is added to the updated **Appendix F.**
>
>
>
> ------
>
>
> **Q2.** From Figure 6(d), we can observe that the following 4 tasks have the largest weights, i.e, EP, AM, IG, and CP. How about we only use these 4 tasks and learning the their weights for prediction?
>
> **A4.**
>
> Figure 6(d) represents only the result of one instance. The tasks with higher weights in this instance may not necessarily have high weights in other instances.
>
> Using only these four tasks could potentially lead to a decline in performance. In the table below, "7-tasks" indicates that the task pool contains all 7 tasks, while "4-tasks" means the task pool includes only four tasks (EP, AM, IG, and CP).
>
> |         | 7-tasks      | 4-tasks      | EP           | AM           | IG           | CP           |
> | :------ | :----------- | :----------- | ------------ | ------------ | ------------ | ------------ |
> | Toxcast | 63.7$\pm$1.5 | 62.8$\pm$1.1 | 59.0$\pm$0.9 | 62.6$\pm$0.1 | 60.7$\pm$0.9 | 61.7$\pm$0.7 |
>
> With the reduction of tasks, there was a decrease in performance, but it still remained better than using a single task.
>
> ---
>
> In light of these responses, we hope we have addressed your concerns, and hope you will consider increasing your score. If we have left any notable points of concern unaddressed, please share and we will attend to these points.

---

> ### Author Response · Authors · 2023-11-21
> **Looking forward to hearing from you**
>
> Dear Reviewer pHBL,
>
> We sincerely thank you for taking the time to review our manuscript and providing valuable suggestions.
>
> Unlike previous years, there will be no second stage of author-reviewer discussions this year, and a recommendation needs to be provided by November 22, 2023. Considering that the author-reviewer discussion phase is nearing the end, we would like to be able to confirm whether our responses have addressed your concerns.
>
> We have provided detailed replies to your concerns a few days ago, and we hope we have satisfactorily addressed your concerns. If so, could you please consider increasing your rating? If you still need any clarification or have any other concerns, please feel free to contact us and we are happy to continue communicating with you.
>
> Best,
>
> Authors

---

### Official Review · Reviewer_C6hu · 2023-11-03

**Soundness:** 3 good
**Presentation:** 3 good
**Contribution:** 2 fair
**Rating:** 5
**Confidence:** 4

**Summary:**

In recent times, graph pre-training for graph representation learning has gained prominence, with numerous pre-training tasks emerging. Integrating knowledge from various pre-training tasks is now a focal research area. Two critical collaborative processes for integration are: (1) selecting the best task combination considering their compatibility and (2) determining the importance of the chosen tasks. While much research has been on weighing tasks, selection has received lesser attention. This paper introduces a new instance-level framework named Weigh And Select (WAS) that merges both processes using decoupled Siamese networks. WAS adaptively determines the best task combination for individual instances, leading to a tailored instance-level task weighing strategy. Experiments across 16 graph datasets reveal WAS's efficacy, producing results comparable to top-performing methods by merging several basic tasks.

**Strengths:**

S1. The authors of the paper have done a good job in providing a compelling motivation for their research. Their thorough analysis of different pre-training tasks combined with a detailed examination of several datasets showcases their comprehensive approach. Furthermore, their assessment of task importance and compatibility provides valuable insights, shedding light on the central issue at hand.

S2. In terms of addressing the research concerns, the authors have carefully identified and highlighted them. These concerns have been formulated into four well-defined research questions that give readers a clear roadmap of the study's objectives. On a broader note, the manuscript is well-organized, with a structured flow that facilitates easy comprehension, making the writing lucid and clear to the audience.

S3. With respect to the empirical aspect of the study, the authors have presented an exhaustive set of experimental results. These results showcase outcomes that are indeed promising.

**Weaknesses:**

W1. The concept of instance is not well defined. Different pre-training tasks may have different definitions of instances fundamentally (eg. at a node-level, subgraph/graph-level, edge-level etc). Having a common and converged definition may not work well. Furthermore, downstream task could also have different definitions of instances.

W2. While 4 questions are clear in the identification of the 4 steps/issues to address, the solutions are quite standard. E.g. using Gumbel-Softmax sampling for Bernoulli distribution,  the weighting scheme and the Siamese network architecture are all well known tools.

W3. The compatibility issue, or interferences among tasks have been observed in previous work in other areas or problem settings [a,b,c]. Some discussion on this aspect, and its particular challenges in graph context, would further strengthen the motivation of the paper.

[a] Zeyuan Wang, Qiang Zhang, HU Shuang-Wei, Haoran Yu, Xurui Jin, Zhichen Gong, and Huajun Chen. 2022. Multi-level Protein Structure Pre-training via Prompt Learning. In The Eleventh International Conference on Learning Representations.
[b] Sen Wu, Hongyang R Zhang, and Christopher Ré. 2020. Understanding and improving information transfer in multi-task learning. arXiv preprint arXiv:2005.00944 (2020).
[c] Tianhe Yu, Saurabh Kumar, Abhishek Gupta, Sergey Levine, Karol Hausman, and Chelsea Finn. 2020. Gradient surgery for multi-task learning. Advances in Neural Information Processing Systems 33 (2020), 5824–5836.

**Questions:**

Please see weaknesses

---

> ### Author Response · Authors · 2023-11-17
> **Response to Reviewer C6hu (1/2)**
>
> Dear Reviewer C6hu:
>
>
> Thank you for your valuable feedback and constructive suggestions. We sincerely appreciate your acknowledgment of compelling motivation, thorough analysis and well-organized manuscript. In the revised manuscript, we have highlighted existing content (which may have been omitted, but which we would like to bring to your attention) in red and revised or new content in blue.
>
> Our detailed response to your concerns is listed as follows:
>
> ------
>
> **W1.** The concept of instance is not well defined. Different pre-training tasks may have different definitions of instances fundamentally (eg. at a node-level, subgraph/graph-level, edge-level etc). Having a common and converged definition may not work well. Furthermore, downstream task could also have different definitions of instances.
>
> **A1.**
>
> We kindly point out that your concerns are the same as those mentioned in our manuscript.
>
> > " The concept of instance is not well defined. "
>
> We provided an example of instances on page 1,Line 36-37 ("e.g., nodes in a social network or graphs in a molecular dataset").
>
> > " Different pre-training tasks may have different definitions of instances fundamentally "
>
>  We completely agree with your statement and we discussed this in Line 191-193 (Original text: "For example, GraphCL performs contrastive learning between instances, while AM masks node attributes on a single instance"). For a graph-level task, GraphCL defines an instance as a graph, whereas for AM, the instance is still a node.
>
> It might be that our expression was not clear enough. We will modify it to "For example, GraphCL performs contrastive learning between graphs, while AM masks node attributes on a single graph".
>
> > "downstream task could also have different definitions of instances."
>
> We also fully agree with this. Therefore, we conducted experiments at both the node-level and graph-level to demonstrate the effectiveness of our method.
>
> **W2.** While 4 questions are clear in the identification of the 4 steps/issues to address, the solutions are quite standard. E.g. using Gumbel-Softmax sampling for Bernoulli distribution, the weighting scheme and the Siamese network architecture are all well known tools.
>
> **A2.**
>
> We sincerely thank you for appreciating our work "4 questions are clear in the identification of the 4 steps/issues to address".
>
> The Gumbel-Softmax, weighing scheme, and the Siamese architecture are seen as a tool rather than a contribution.
>
> Our contributions are:
>
> 1. examining the shortcomings of various existing multi-task graph learning methods (they mainly based on weighing-only method), i.e., not considering the compatibility issue.
> 2. distinguishing between the compatibility issue and the importance issue, and using decoupling frameworks to deal with these two issues.
>
> Considering that we are already achieving what the framework was designed for (1. good performance 2. the ability to sustain performance growth) by using those classic components, we kindly point out that we think that there is no need to force modifications to the classic components just to appear that we are different.

---

> ### Author Response · Authors · 2023-11-17
> **Response to Reviewer C6hu (2/2)**
>
> **W3.** The compatibility issue, or interferences among tasks have been observed in previous work in other areas or problem settings. Some discussion on this aspect, and its particular challenges in graph context, would further strengthen the motivation of the paper.
>
> **A3.**
>
> We sincerely appreciate your suggestions for enhancing the motivation of our manuscript, which will make it stronger! Among the works you mentioned\[a]\[b]\[c], \[a] is specifically designed for transformers. However, the dominant architecture in the graph domain is still GNN, which limits its applicability.
>
> \[b] differs in focus from our method. We focus on extracting and combining knowledge from various pre-trained tasks to enhance the performance of downstream tasks, while \[b] is concerned with handling different downstream tasks when using a single model. \[c] combines tasks through gradient surgery, a method similar to AutoSSL and ParetoGNN mentioned in our paper. These approaches combine task losses through various weighting methods. Firstly, they are unable to provide a customized combination for each instance. Secondly, they overlook compatibility issues, making it difficult to achieve continual performance improvement as the task pool expands.  Moreover, these works do not address the concept of "compatibility". Their solutions differ from ours as their main focus remains on various weighting methods, overlooking the necessity of selection.
>
> We have updated Sec.2 to include these more discussions.
>
> ---
>
> [a] Zeyuan Wang, Qiang Zhang, HU Shuang-Wei, Haoran Yu, Xurui Jin, Zhichen Gong, and Huajun Chen. 2022. Multi-level Protein Structure Pre-training via Prompt Learning. In The Eleventh International Conference on Learning Representations.
>
>  [b] Sen Wu, Hongyang R Zhang, and Christopher Ré. 2020. Understanding and improving information transfer in multi-task learning. arXiv preprint arXiv:2005.00944 (2020).
>
>  [c] Tianhe Yu, Saurabh Kumar, Abhishek Gupta, Sergey Levine, Karol Hausman, and Chelsea Finn. 2020. Gradient surgery for multi-task learning. Advances in Neural Information Processing Systems 33 (2020), 5824–5836.
>
> ---
>
> In light of these responses, we hope we have addressed your concerns, and hope you will consider increasing your score. If we have left any notable points of concern unaddressed, please share and we will attend to these points.

---

> > ### Comment · Reviewer_C6hu · 2023-11-18
> >
> > Thank you for the detailed responses. I will weigh them carefully in the final review.

---

> > > ### Comment · Reviewer_C6hu · 2023-12-03
> > >
> > > I still have concern about W2 and W3. In W3, while [a] is designed for transformer, its idea can be extended to GNNs. It would be cursory to disregard it just because it is originally not designed for GNNs. Although I agree that [a] would not fully address the compatibility issue as well.
> > >
> > > I would keep my score (marginally below acceptance), although I wouldn't mind if the paper is accepted.

---

> ### Author Response · Authors · 2023-11-19
> **Replying to Reviewer C6hu**
>
> Dear Reviewer C6hu,
>
> We sincerely thank you for taking the time to review our manuscript and providing valuable suggestions.
>
> Unlike previous years, there will be no second stage of author-reviewer discussions this year, and a recommendation needs to be provided by November 22, 2023. We hope we have satisfactorily addressed your concerns. If so, could you please consider increasing your rating? If you still need any clarification or have any other concerns, please feel free to contact us and we are happy to continue communicating with you.
>
> Best,
>
> Authors

---

### Official Review · Reviewer_fcHp · 2023-11-05

**Soundness:** 3 good
**Presentation:** 4 excellent
**Contribution:** 3 good
**Rating:** 8
**Confidence:** 3

**Summary:**

Since the roles of pre-training tasks vary with downstream tasks and different pre-training tasks may not be compatible, the authors point out that selecting and weighting tasks are key parts of the pre-training. Moreover, the existing methods that select tasks based on learned weights would confuse the roles of selection and weighting. To address these limitations, they propose WAS to decouple these two processes.

**Strengths:**

1. The proposed WAS decouples the selecting and weighting to avoid performance reduction caused by task conflicts. They calculate weights based on output distributions instead of losses to address the non-comparability between different loss functions.
2. The proposed method can automatically select the number and type of suitable pre-training tasks for different downstream instances.
3. The authors conduct extensive comparison experiments, and the proposed WAS performs better on both node-level and graph-level tasks. In addition, they visualized the evolution processes of selecting tasks and updating weights, which proves the decoupling of selection and importance weighting.

**Weaknesses:**

1. The authors should add more ablation experiments to demonstrate the effectiveness of the proposed model, such as the role of different updating methods in decoupling the selecting and weighting outputs, and the role of reweighting after selection.
2. The comparison model in A5 needs more description to distinguish it from A1, such as whether Random-Select and ALL use learned weights.

**Questions:**

Please refer to Weaknesses.

**Details Of Ethics Concerns:**

No ethical issues.

---

> ### Author Response · Authors · 2023-11-17
> **Response to Reviewer fcHp**
>
> Dear Reviewer fcHp:
>
>
> Thank you for your valuable feedback and constructive suggestions. We sincerely appreciate your acknowledgment of the effectiveness of our work. In the revised manuscript, we have highlighted existing content (which may have been omitted, but which we would like to bring to your attention) in red and revised or new content in blue. Our detailed response to your concerns is listed as follows:
>
> ------
>
> **W1.** The authors should add more ablation experiments to demonstrate the effectiveness of the proposed model, such as the role of different updating methods in decoupling the selecting and weighting outputs, and the role of reweighting after selection.
>
> **A1.**
>
> You mentioned "the role of different updating methods", and we speculate that you hope to further understand the relationship between the momentum-update method and the decoupling of the selecting and weighting modules. It's about our design intuition, as we want the result of selecting to utilize the information from the importance, not the other way around. Considering that a traditional ablation (removing the momentum update method) would render our method completely ineffective, we use a comparative approach. In the table below, Method A represents gradient-based updating for weighing and momentum-based updating for selecting (i.e., same as WAS); Method B represents gradient-based updating for selecting and momentum-based updating for weighing. It can be seen that Method B performs much worse than Method A (WAS). This indicates that our update method is more effective and aligns better with the design concept.
>
> |            | BACE         | BBBP         | **Sider**    | **HIV**      |
> | :--------- | :----------- | :----------- | :----------- | :----------- |
> | *Method A* | 80.7$\pm$0.5 | 70.5$\pm$1.0 | 60.4$\pm$0.4 | 78.4$\pm$1.9 |
> | *Method B* | 73.1$\pm$0.9 | 62.8$\pm$1.5 | 51.8$\pm$3.2 | 73.4$\pm$1.8 |
>
> Regarding "the role of reweighting after selection," this is done to standardize the magnitude of knowledge distilled to the student model. For instance, if we select teachers with weights of 0.1 and 0.2 from three teachers weighted as 0.1, 0.2, and 0.7, but do not perform reweighting, the final sum of the weights will not equal 1. This would result in the number of selected teachers significantly impacting performance. As can be seen from the experiment, eliminating this would make the results very poor, as the teacher would be completely unable to guide the students.
>
> |                       | BACE         | HIV          | **Tox21**    | **BBBP**     |
> | :-------------------- | :----------- | :----------- | :----------- | :----------- |
> | *WAS*                 | 80.7$\pm$0.5 | 78.4$\pm$1.9 | 75.4$\pm$0.7 | 70.5$\pm$1.0 |
> | *WAS w/o re-weighing* | 65.1$\pm$3.7 | 63.3$\pm$2.4 | 64.1$\pm$3.0 | 60.7$\pm$5.1 |
>
> This additional result has already been added to the Table.5. Thank you again for your suggestion!
>
>
>
> **W2.** The comparison model in A5 needs more description to distinguish it from A1, such as whether Random-Select and ALL use learned weights.
>
> **A2.**
>
> We sincerely appreciate your suggestions for enhancing the readability and clarity of our manuscript, which will make it stronger!
>
> Both Random-Select and ALL use learned weights, but Random-Select randomly select teachers, and ALL select all the teachers. We add corresponding explanations in the updated manuscript to better clarify this point.
>
> ---
>
> In light of these responses, we hope we have addressed your concerns. If we have left any notable points of concern unaddressed, please share and we will attend to these points.

---

> ### Author Response · Authors · 2023-11-21
> **Looking forward to hearing from you**
>
> Dear Reviewer fcHp,
>
> We sincerely thank you for taking the time to review our manuscript and providing valuable suggestions.
>
> Unlike previous years, there will be no second stage of author-reviewer discussions this year, and a recommendation needs to be provided by November 22, 2023. Considering that the author-reviewer discussion phase is nearing the end, we would like to be able to confirm whether our responses have addressed your concerns.
>
> We have provided detailed replies to your concerns a few days ago, and we hope we have satisfactorily addressed your concerns. If you still need any clarification or have any other concerns, please feel free to contact us and we are happy to continue communicating with you.
>
> Best,
>
> Authors

---

> > ### Comment · Reviewer_fcHp · 2023-11-22
> > **Thank you for the responses**
> >
> > I have read the author's response and other reviews. I would like to keep my review score for the paper and encourage the authors to incorporate these discussions into the main paper.

---

### Author Response · Authors · 2023-11-17
**Global responses**

We are grateful to all reviewers for their time and their constructive suggestions, which we agree will significantly improve the communication of our work.

We are very encouraged by reviewers’ positive evaluation on the quality of this work, "conduct extensive comparison experiments" (**fcHp**), "have done a good job" (**C6hu**), "clearly written and easy to follow" (**pHBL**), "new and insightful" (**66Ho**).

Here is a description of the revisions we made to our manuscript. In the revised manuscript, we have highlighted existing content (which may have been omitted, but which we would like to bring to your attention)  in red and revised or new content in blue.

1. Line 103-109. We have added more discussion on related work.
2. Line 191-193. We have modified the sentence to highlight more clearly that "the definition of instances may vary across pre-training tasks".
3. Line 344-348 & Table.5. We have added ablation experiments on re-weighing and a description of the relevant.
4. Line 650-656. We added more experiments to show the number of teachers selected on different datasets and that the selecting module does not necessarily select the teacher with the highest weight.
5. Line 656-684. We have added a section of experiments and theory to further explain why the selecting module is essential.

---

### Author Response · Authors · 2023-11-20
**Looking forward to your feedback**

Dear Reviewers,

We sincerely thank you for taking the time to review our manuscript and providing valuable suggestions.

Unlike previous years, there will be no second stage of author-reviewer discussions this year, and a recommendation needs to be provided by November 22, 2023.  Considering that the author-reviewer discussion phase is nearing the end, we would like to be able to confirm whether our responses have addressed your concerns.

We have provided detailed replies to your concerns a few days ago, and we hope we have satisfactorily addressed your concerns. If you still need any clarification or have any other concerns, please feel free to contact us and we are happy to continue communicating with you.



Best,

Authors

---

> ### Author Response · Authors · 2023-11-22
>
> Dear Reviewers,
>
> We have updated the revised version of our manuscript. The new version includes more experiments, analyses, theories, and explanations to address your concerns. If you have any concerns that have not been resolved, please let us know and we are happy to continue communicating with you.
>
> Best,
>
> Authors

---

### Meta-Review · Area_Chair_fCbB · 2023-12-14

**Metareview:**

The paper studies how to make use of different graph pre-training tasks to improve performance for downstream task during fine-tuning. The paper proposed an instance-level framework called Weigh And Select (WAS) that determines the best pre-training task combination per instance during fine-tuning. The method showed improved performance against baseline methods and multi-task learning approaches.

Strength: good motivation and extensive experiments.
Weakness: the proposal combines several standard and well-known techniques for the problem, e.g. Gumbel-Softmax sampling, Siamese networks, thus the new contribution is limited. The authors performed complexity analyses, but the results were not carefully analyzed to understand if the quality gain justified the added cost. The reviewers remain lukewarm after rebuttal.

**Justification For Why Not Higher Score:**

The proposal combines several standard and well-known techniques for the problem, e.g. Gumbel-Softmax sampling, Siamese networks, thus the new contribution is limited. The authors performed complexity analyses, but the results were not carefully analyzed to understand if the quality gain justified the added cost.

**Justification For Why Not Lower Score:**

The paper is well motivated, and the proposed method is simple and seems effective from empirical evaluations.

---

### Decision · Program_Chairs · 2024-01-16

Accept (poster)